# Single-cell analysis of the aged ovarian immune system reveals a shift towards adaptive immunity and attenuated cell function

Tal Ben Yaakov[1], Tanya Wasserman[1], Eliel Aknin[1], Yonatan Savir[1,2]*

[1]Department of Physiology, Biophysics and Systems Biology, Rappaport Faculty of Medicine, Technion – Israel Institute of Technology, Haifa, Israel; [2]Rappaport Family Institute for Research in the Medical Sciences, Haifa, Israel

**Abstract** The immune system plays a major role in maintaining many physiological processes in the reproductive system. However, a complete characterization of the immune milieu in the ovary, and particularly how it is affected by female aging, is still lacking. Here, we utilize single-cell RNA sequencing and flow cytometry to construct the complete description of the murine ovarian immune system. We show that the composition of the immune cells undergoes an extensive shift with age towards adaptive immunity. We analyze the effect of aging on gene expression and chemokine and cytokine networks and show an overall decreased expression of inflammatory mediators together with an increased expression of senescent cells recognition receptors. Our results suggest that the fertile female's ovarian immune aging differs from the suggested female post-menopause inflammaging as it copes with the inflammatory stimulations during repeated cycles and the increasing need for clearance of accumulating atretic follicles.

## Editor's evaluation

The study describes a single-cell analysis of the mammalian ovary in young, adult, and old mice, and is an important contribution to the field identifying clusters of immune cell populations across the different ages. The combination of single-cell RNA sequencing and flow cytometry used is a robust and unbiased approach that provides compelling evidence of immune cell alterations in aged ovaries.

## Introduction

One of the effects of aging in mammalians is a decline in fertility and hence a diminished capability to give birth to offspring (*Pal and Santoro, 2003*). In women from their early 30's, there is a sharp decrease in fertility, accompanied by an exponentially increase in the odds of miscarriages and birth defects, alongside a drastically lower success rate of in-vitro fertilization (IVF) procedure (*Madankumar et al., 2003*; *Nelson and Lawlor, 2011*; *Szamatowicz and Grochowski, 1998*). Another well-documented effect of age is on the ability of the immune system to overcome illnesses and eliminate different pathogens (*Kovacs et al., 2009*; *Plowden et al., 2004*; *Solana et al., 2006*; *Weiskopf et al., 2009*).

The presence of immune cells such as macrophages (Mφs), dendritic cells (DCs), granulocytes, T and B lymphocytes was identified through the entire female reproductive tract (*Givan et al., 1997*; *Lee et al., 2015*), and in the ovaries in particular (*Best et al., 1996*; *Bukulmez and Arici, 2000*;

*For correspondence:
yoni.savir@technion.ac.il

**Competing interest:** The authors declare that no competing interests exist.

*Carlock et al., 2013*; *Yang et al., 2019*). These cells participate in many fertility-related processes in the ovaries – from follicle development up to ovulation and corpus luteum formation and regression (*Cohen-Fredarow et al., 2014*; *Fair, 2015*; *Oakley et al., 2010*; *Wu et al., 2004*; *Yang et al., 2019*). Ovulation, for example, is considered an inflammatory process that includes edema, vasodilation, pain, and heat (*Duffy et al., 2019*; *Richards et al., 2008*). Changes in the immune milieu, such as depletion of Mφs and DCs have been shown to result in a decreased number of ovulated oocytes, depletion of endothelial cells, increased follicular atresia, and lead to a delayed progression of the estrus cycle (*Cohen-Fredarow et al., 2014*; *Turner et al., 2011*; *Wu et al., 2004*).

Characterizing the complete immune milieu in the ovary, and in particular how it is affected by female aging, is challenging mainly due to the small fraction of the immune cells compared to other cell types in the ovary (*Lliberos et al., 2021*; *Wagner et al., 2020*). For that reason, previous work using whole-ovary single-cell measurements did manage to portray the ovary's main cell types (oocytes, granulosa, theca, immune, etc.) yet didn't have the resolution to resolve the entire immune milieu (*Fan et al., 2019*; *Lliberos et al., 2021*; *Wagner et al., 2020*). Others have a priori focused on a limited set of cell types (*Cohen-Fredarow et al., 2014*; *Wu et al., 2004*), while additional studies have used bulk RNA sequencing experiments (*Ma et al., 2020*), hence did not capture the entire immune members in the ovaries. In this work, we were able to isolate the immune cells and perform consecutive single-cell analyses. Investigating the age effect on the immune system in the single-cell level have also been done in several recent studies (*Almanzar et al., 2020*; *Kimmel et al., 2019*; *Mogilenko et al., 2021*). However, their focus was on other non-fertility related tissues such as bladder, kidney, peritoneum and more. In addition, the aged group was very old, far past estropause. Attempts to characterize the effects of female age on the immune system in the ovary were limited to a small subset of cells (*Lliberos et al., 2021*; *Zhang et al., 2020*), addressing mainly changes in the macrophages fraction that decreases with age, and accumulation of inflammatory mediators such as cytokines and reactive oxygen species within the tissue (*Lliberos et al., 2021*; *Zhang et al., 2020*).

In this work, we provide the first complete detailed characterization of the murine ovarian immune system composition at the single-cell level. We show the presence of various immune cell populations, such as Mφs, DC's, neutrophils (NTs), NK cells, NKT cells, innate lymphoid cells (ILCs), B cells, and several T cell types – including an ovary specific CD3$^+$ CD4$^-$ CD8$^-$ double-negative T (DNT) cells. Moreover, we show an extensive tissue-specific effect of female age on the ovarian immune milieu, resulting in a shift towards adaptive immunity, mainly by a significant increase in the DNT population. In addition, we analyzed the changes in gene expression of the cells and discovered a global attenuation in their general function and responsiveness. We also found a decrease in the expression of inflammatory mediators such as cytokines and chemokines. Moreover, we identified some evidence for an increase in senescent cells recognition activity. Our results serve as an opening for a much more comprehensive understanding of the interaction between female aging and the immune system in fertile female mammals.

## Results

### The ovarian immune milieu is altered with age

To characterize the ovarian immune milieu, we have isolated immune cells (CD45$^+$ cells) from the ovaries of young (11–15 weeks), adult (20–37 weeks), and old (40–47 weeks) virgin mice, and utilized flow cytometry and single-cell RNA sequencing (scRNA-seq) to characterize the ovarian immune cells and how they are affected by female age (*Figure 1A*). First, we performed scRNA-seq on isolated immune cells from the ovaries of 13 weeks old mice (n=2; 3307 cells). To cluster the cells and identify their type, we used a combination of both literature-based annotation and automatic annotation methods (Seurat R package and SingleR algorithm, "Methods" and *Figure 1—figure supplement 1*). In addition, we performed a batch correction analysis to validate the clusters that emerged from the tSNE analysis (*Figure 1—figure supplement 2*). The combination of these methods allowed us to identify within the ovaries the following cell types: Mφs, DCs, NTs, B cells, NK cells, NKT cells, ILC1, ILC2, ILC3, and several clusters of T lymphocytes: CD8$^+$ (CD8 T), CD4$^+$ (CD4 T), and CD4$^-$ CD8$^-$ double-negative T cells (DNT cells) (*Figure 1B and C* and *Figure 1—figure supplement 3*). Most of the cells were innate immune cells, mostly ILC1, Mφs, NTs, and NK cells.

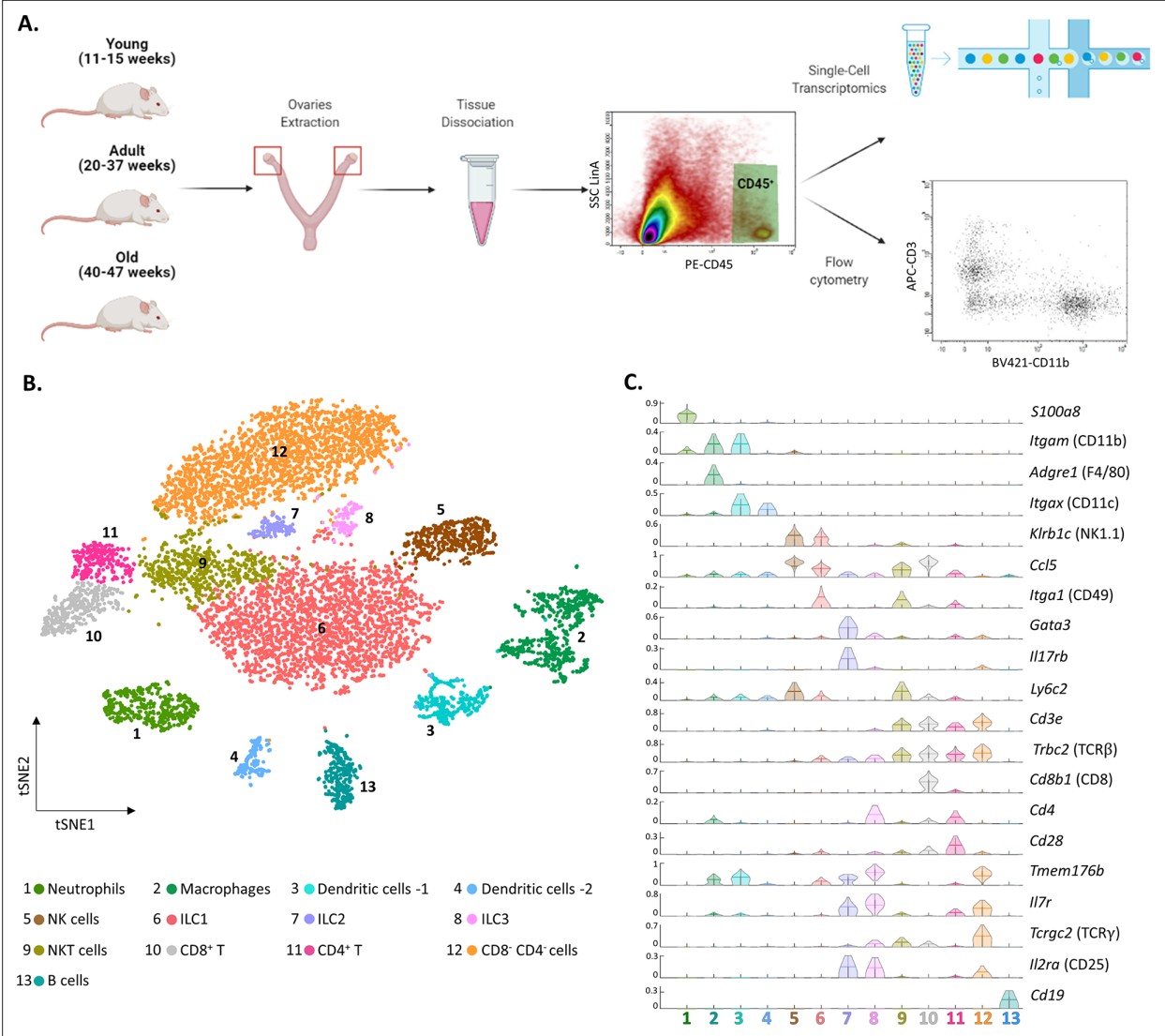

**Figure 1.** The ovarian immune milieu is consisted of various cell types. (**A**) Schematic illustration of the experimental pipeline (created with BioRender. com). Ovaries of female mice at different ages were extracted. Then, cells were gated for CD45 expression and further analyzed using single-cell RNA sequencing or flow cytometry. (**B**) tSNE plot of joint data from both samples (young and old), divided into clusters. (**C**) Violin plot of normalized expression for cluster-specific markers. Each row represents the normalized expression of a single marker across all immune clusters. Normalized expression values are between 0–1.

The online version of this article includes the following source data and figure supplement(s) for figure 1:

**Source data 1.** Young raw data.

**Figure supplement 1.** Automatic ovarian immune cells annotation.

**Figure supplement 2.** The effect of batch correction.

**Figure supplement 3.** Literature-based annotation of ovarian immune cell populations.

**Figure supplement 4.** Flow cytometry gating strategy.

**Figure supplement 5.** Flow cytometry measurement of Group 1 innate lymphoid cells (G1-ILCs) fraction in the ovaries.

**Figure supplement 6.** Dendritic cells clusters are composed from cDC1 and cDC2 populations.

**Figure supplement 7.** Double-negative T cells are more abundant in the ovaries compared to other tissues.

**Figure supplement 8.** Ovarian CD3+ CD4- CD8- cells are predominantly TCRγδ-.

To validate the presence of the immune populations that emerged from our scRNA-seq experiments, we used flow cytometry (the gating strategy is shown in *Figure 1—figure supplement 4*). First, we measured the fractions of group 1 of innate lymphoid cells (G1-ILC), CD45[+] NK1.1[+] CD3[-] cells (i.e. NK and ILC1) in the mouse ovaries and spleen (*Figure 1—figure supplement 5*). The average fraction of G1-ILC in the spleen was 11.81% and in the ovary was 42.71%. These results are consistent with the high G1-ILC fraction resulting from the ovary scRNA-seq analysis (35.4% ILC1 and 9.8% NK), and previous results, which demonstrated that G1-ILC proportion in mice spleen is relatively low (*Boulenouar et al., 2017*). In addition, further characterization of the DCs clusters (3 and 4) revealed that their transcriptomic signature corresponds to conventional dendritic cells type 2 (cDC2) and type 1 (cDC1), respectively (*Figure 1—figure supplement 6*). Among other cell types that were found, DNT cells are unique, somewhat less well-defined cell population.

To confirm the presence of CD4[-] CD8[-] T cells in the ovaries, we conducted a flow cytometry experiment comparing the fractions of CD4[+], CD8[+], and CD4[-] CD8[-] T cells in the mouse ovaries, spleen, and peritoneum (*Figure 1—figure supplement 7*). Using additional flow cytometry experiments we validated that ovarian DNT cells are TCRγδ[-] (*Figure 1—figure supplement 8*). These measurements

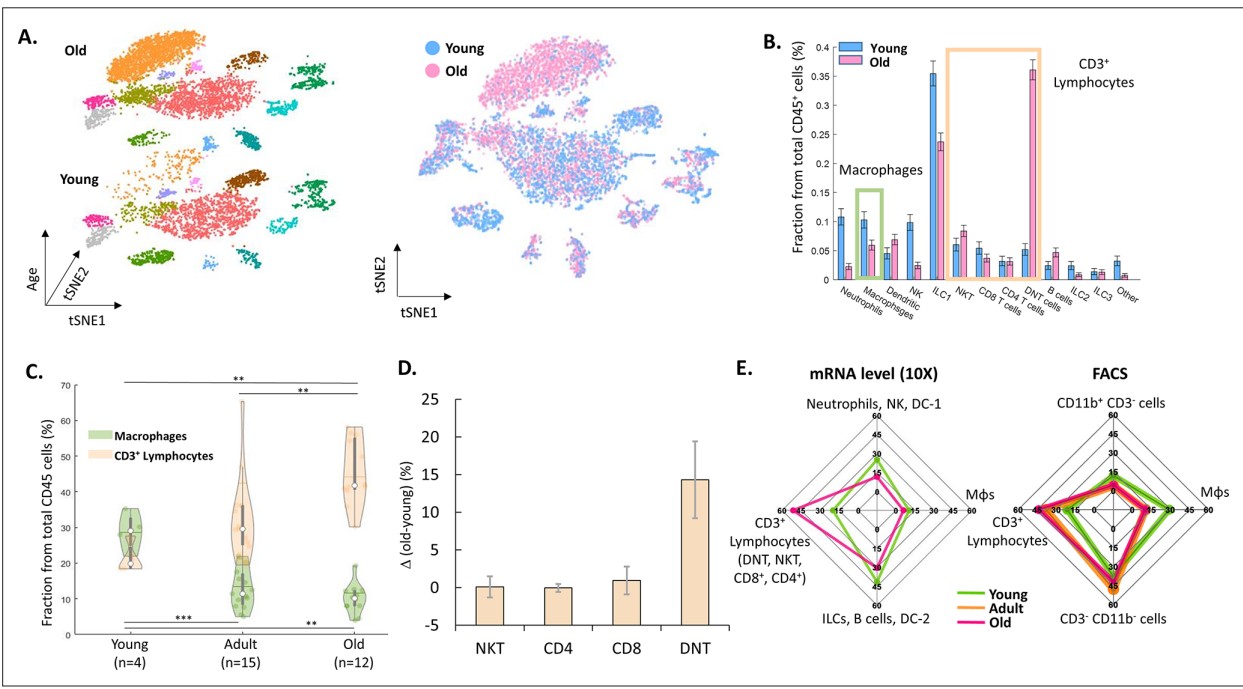

**Figure 2.** The effect of female age on the ovarian immune milieu. (**A**) 3-D tSNE plot (left) and an overlay (right) of all ovarian CD45[+] cells found in scRNA-seq, divided by age group. (**B**) The effect of female age on the fractions of each cell type, with a confidence interval of 95% at the top of each bar. The green and yellow rectangles mark the macrophages and CD3[+] populations, respectively. (**C**) Violin plot of the changes in fraction distributions of macrophages and CD3[+] lymphocytes as a function of age as measured by flow cytometry (Kolmogorov-Smirnov test, ** p-value <10[−2], *** p-value <10[−3]). (**D**) Change in the fraction of different CD3[+] population comparing old (for CD4 and CD8 T cells – 42.6–49.6 weeks, n=4; for DNT and NKT cells – 49.6 weeks, n=2) and young (for CD4 and CD8 T cells – 10.1–14.5 weeks, n=5; for DNT and NKT cells – 10.1 weeks, n=3) mice as measured using flow cytometry. Error bars denote standard deviation. (**E**) Comparison between transcriptome and protein level of immune populations within the ovaries at different female ages. Each spider plot shows the distribution of different immune cell types measured using scRNA-seq (left panel) and flow cytometry (right).

The online version of this article includes the following source data and figure supplement(s) for figure 2:

**Source data 1.** Old raw data.

**Figure supplement 1.** Significant changes in cell type abundance at old age.

**Figure supplement 2.** Flow cytometry measurements of additional ovarian immune populations.

**Figure supplement 3.** Changes in ovarian CD3[+] cells and macrophages.

**Figure supplement 4.** Splenic CD3[+] cells fraction is decreasing at older age.

**Figure supplement 5.** Changes in ovarian CD3[+] cells and Macrophages is cycle stage independent.

validate the scRNA-seq results and show that although present in other tissues at small fractions, CD3$^+$ TCRβ$^+$ CD4$^-$ CD8$^-$ cells are tissue-specific cells to the ovaries.

Next, we examined the changes in the ovarian immune milieu at older ages. Using cells isolated from old, near estropause mouse (43 weeks; the rodent equivalent of the human menopause; 5468 cells), we characterized the old ovarian immune system (using the same annotation methods) and compared it to its younger counterpart (*Figure 2A and B*). The results demonstrated a shift at older age towards a lymphocytes-rich environment that was accompanied by decreased fractions of several immune populations such as ILC1 cells, Mφs, NTs, and NK cells (*Figure 2B*, *Figure 2—figure supplements 1–2*). To both validate the scRNA-seq results and to check whether this effect is cycle-stage dependent, we conducted several flow cytometry experiments.

To measure the effect of female age on the fractions of T-lymphocytes (CD3$^+$) and Mφs (CD11b$^+$ F4/80$^+$) from total ovarian CD45$^+$ cells, mice were divided into three groups of age: young (11–15 weeks, n=4), adult (20–37 weeks, n=15) and old (40–47 weeks, n=12). The results show a significant increase with age of CD3$^+$ cells' fraction (young vs. adult and adult vs. old, Kolmogorov-Smirnov test, p-value $<10^{-2}$), while the fraction of Mφs was significantly decreased (young vs. adult and young vs. old, Kolmogorov-Smirnov test, p-value $<10^{-3}$ and p-value $<10^{-2}$, respectively) at older ages. Moreover, flow cytometry experiments validated that, as with the scRNA-seq results, most of the change in CD3$^+$ lymphocytes' fraction is due to a substantial increase in DNT cells at old age (*Figure 2C, D and E* and *Figure 2—figure supplement 3*). Furthermore, analysis of splenic CD3$^+$ lymphocytes show that in contrast to the ovaries, the fraction of these cells decreases at old age, while the fraction of DNT cells doesn't change (*Figure 2—figure supplement 4*). These age-dependent results are cycle-stage independent (*Figure 2—figure supplement 5*). Taking the results both from the scRNA-seq and the flow cytometry (*Figure 2E*), there is a consistent shift towards adaptive immunity (an increase in the CD3$^+$ lymphocytes fraction), while most innate immune cells' fraction (Mφs, NKs, ILC1, and NTs) decreases.

## The female aging effect on the ovarian immune cells' transcriptome

After identifying the ovarian immune milieu and the changes it undergoes at older age, we characterized the changes in gene expression within each immune cluster (*Supplementary file 1*). *Figure 3A* depicts the differentially expressed genes patterns across all clusters. Several clusters, such as DNT cells, ILC1, NKT cells, and CD4 T cells exhibited an extremely skewed pattern, in which most of their differentially expressed genes (DEGs) were downregulated, compared to upregulated DEGs. This may imply that these cell types are more susceptible to female aging. After defining DEGs for each cell type, we explored changes in biological processes via GO enrichment analysis (*Figure 3B*, 'Methods'). Most cell types showed an enriched set of processes that were downregulated with female age. Using the REVIGO platform ('Methods'), we eliminated redundant GO terms and counted the appearances of each GO term across all cell types (*Supplementary file 2*). Terms that were found in more than one cell type were classified as 'global', and cell type-unique terms as 'specific'. For further analysis, we took all the global terms and used the REVIGO platform to cluster them according to their semantics distance (*Figure 3B*). The results show a global decrease in several clusters of processes. The clusters with the processes that were mostly shared among cell types included decreased biosynthesis and metabolism-related processes. Another distinct cluster includes a general loss of regulation over various processes. The third well-defined cluster includes a decrease in the cellular response to different stimuli.

Among the cell-type-specific downregulated processes, Mφs exhibit attenuation in immune and inflammatory responses, along with decreased tissue remodeling and wound healing processes. DCs show a decrease in cell activation and regulation of immune response. Several immune cell types showed a limited set of enriched upregulated processes, in which the most prominent ones were exhibited by Mφs and DNT cells and included T cell activation and differentiation processes (*Supplementary file 3*).

## Aging affects cytokines and chemokines connectome of ovarian immune cells

To get a better notion of the effect of aging on the different immune cell types, we estimated the effect of aging on the chemokine and cytokines interactions between the immune cells (*Figure 4*). For both chemokines and cytokines, we used the KEGG database to extract the network of ligands

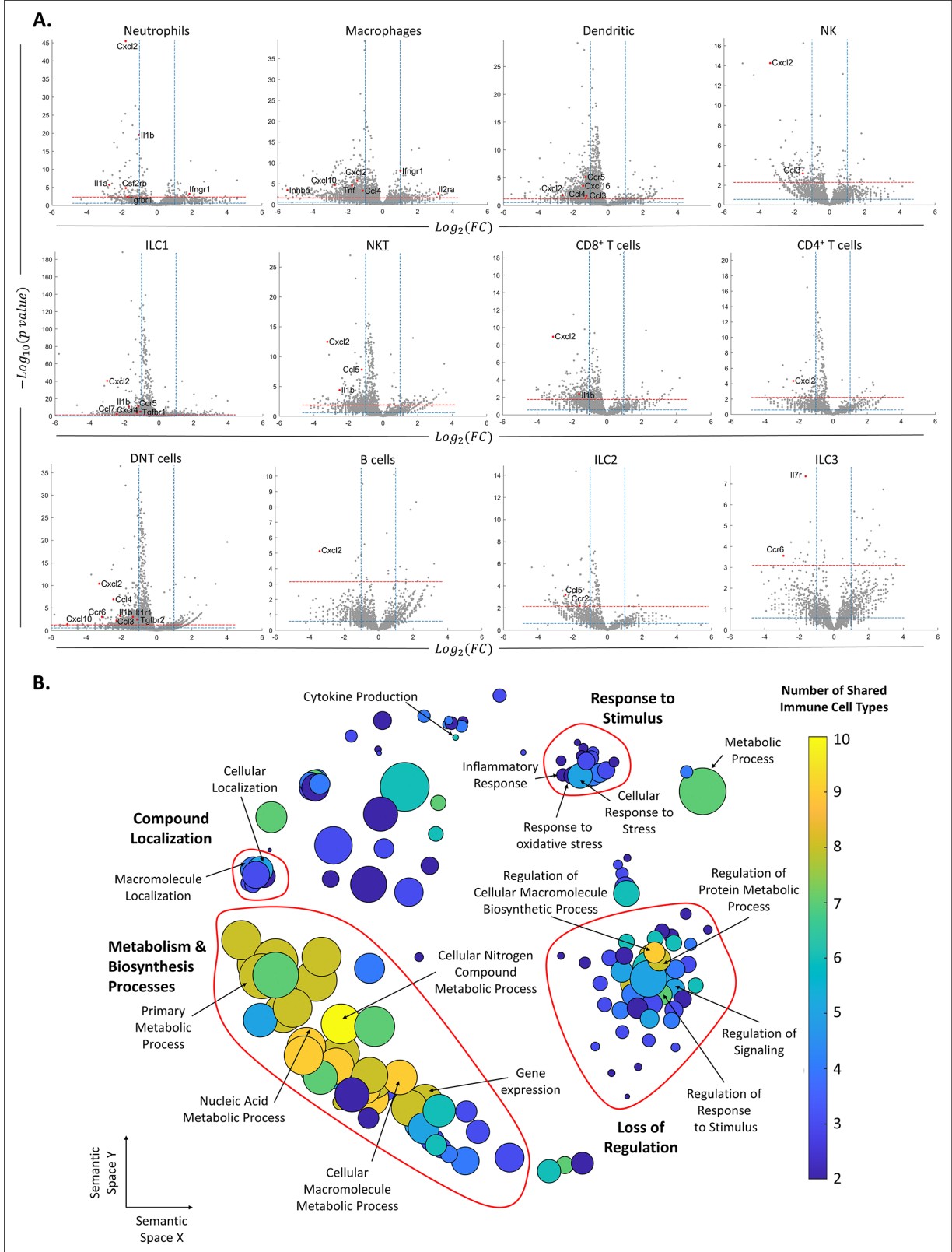

**Figure 3.** Changes in gene expression along with female age. (**A**) Volcano plots of the scRNA-seq analysis for the different immune cell types. The vertical dashed lines mark twofold upregulation and downregulation, the blue horizontal dashed line marks p-value = 0.05, and the red horizontal dashed line marks FDR = 0.1. Genes in the top left section of each graph are significantly downregulated at old age, while genes in the top right section are upregulated at old age. Each grey dot represents the change in the expression of different genes. Red dots denote significantly changed

*Figure 3 continued*

chemokines and cytokines. (**B**) Each circle represents a downregulated biological process in the old mice, which appeared in at least two types of immune cells. The axes represent semantic similarities distance as calculated by REVIGO ('Methods'). The color of each term represents the number of immune cell types in which the process is downregulated with age. The size of each term represents the hierarchy of the biological process; the bigger the circle, the higher the hierarchy of the process is.

and receptors and their interactions ('Methods'). Within each network, we focused on significantly changed connections, which we defined as edges that both of their nodes (i.e. both the ligand and receptor) have been significantly decreased or increased in their expression at old age. Significant nodes can be in the same cell type, or each one in different cell type, and in at least one cell type. We found that almost all significant nodes and edges in both networks were downregulated (*Figure 4C and D*).

The most prominent affected edges in the chemokines network involved *Ccr5* expressed by DCs and ILC1 cells, *Ccr2* expressed by ILC2 cells, and their ligands *Ccl2*, *Ccl3*, *Ccl4*, *Ccl5*, *Ccl7*, *Ccl8*, and *Ccl12* (*Figure 4C*). Both of these receptors have been shown to take part in chemoattraction of immune cells in the context of various inflammatory processes (*D'Ambrosio et al., 2003*; *Mencarelli et al., 2016*; *Proudfoot, 2002*). For further validation, we applied the *cell2cell* algorithm (*Armingol et al., 2022*), which also pointed out *Ccr5* as the main chemokine receptor modulated by age (*Figure 4—figure supplement 1A*). Thus, we measured experimentally the fraction of cells that express CCR5 using flow cytometry experiments, and show it is indeed decreases significantly (*Figure 4—figure supplement 1B*). Moreover, *Cxcl2,* an inflammatory chemokine that mediates neutrophils trafficking (*Lentini et al., 2020*; *Li et al., 2016*), was significantly decreased in almost all immune cell types (*Figure 4A*). CCRL2 is an atypical chemokine receptor that was found to be upregulated in activated immune cells after induction of inflammatory signals (*Del Prete et al., 2013*). *Ccrl2* was downregulated in several cell types such as ILC1 and NKT, although it is mainly expressed by myeloid cells such as NTs, DCs, and Mφs (*Del Prete et al., 2013*).

The changes in the cytokines network were found mainly in the IL-1 superfamily (*Il1r1* and *Il1r2*, along with *Il1a*, *Il1b* and *Il1rn*). Moreover, TNF-receptor *Tnfsfr1b* and its ligand *Tnf* (TNFα) also showed a significant decrease at old age (*Figure 4A*). As IL-1 and TNF superfamily members are considered inflammatory, along with the evident decrease in various inflammatory chemokines and their receptors, our results suggest that beyond overall inhibition in the ovarian immune function, aging also shifts its phenotype towards a less inflammatory state.

To account for possible global downregulation, as emerges from *Figure 3A*, that may lead to an artifact in which DEGs that were found, are not significant under the global downregulation, we've performed another analysis. In this analysis, we considered only the top down/up-regulated genes with a p-value≤0.025, defined by cumulative probability distribution (CDF) analysis. GO enrichment analysis using the new DEGs show that 'Inflammatory response' (GO:0006954) was downregulated in four different clusters (NTs, Mφs, DCs and ILC3 cells). In addition, the process 'Negative regulation of neuroinflammatory response' (GO:0150079) was upregulated in NK cells (see *Supplementary file 4*).

Another evident downregulated edges in the cytokine network were in the TGFβ superfamily (*Gdf11* and *Inhba* along with their receptors *Acvr2a* and *Tgfbr1*). Activin A, a dimer composed of two Inhibin-βA subunits (the translation product of *Inhba*), is produced among others by the gonads and promotes LH secretion from the pituitary. It plays an important role in expanding the primordial follicle pool and contributes to the early stages of follicular growth by increasing FSH receptor expression on granulosa cells (*Namwanje and Brown, 2016*). In addition, Activin A was found to activate resting macrophages – yet there are contradictory findings as to rather its effect is pro or anti-inflammatory (*Morianos et al., 2019*). Decreased expression of both *Inhba* and *Acvr2a* by ovarian macrophages at older age might suggest a specific role of macrophages in supporting follicular growth (via Activin secretion) during the estrous cycle, which decays as age progresses. Moreover, these results may present a mechanism in which macrophages are participating in inducing an inflammatory environment as part of the ovulation process as a response to Activin.

## Aging affects recognition of senescent cells by ovarian immune cells

Inducing cell senescence, which is an irreversible state of growth arrest, is a mechanism the body uses to handle cell stress which can accumulate during aging (*Campisi and d'Adda di Fagagna, 2007*)

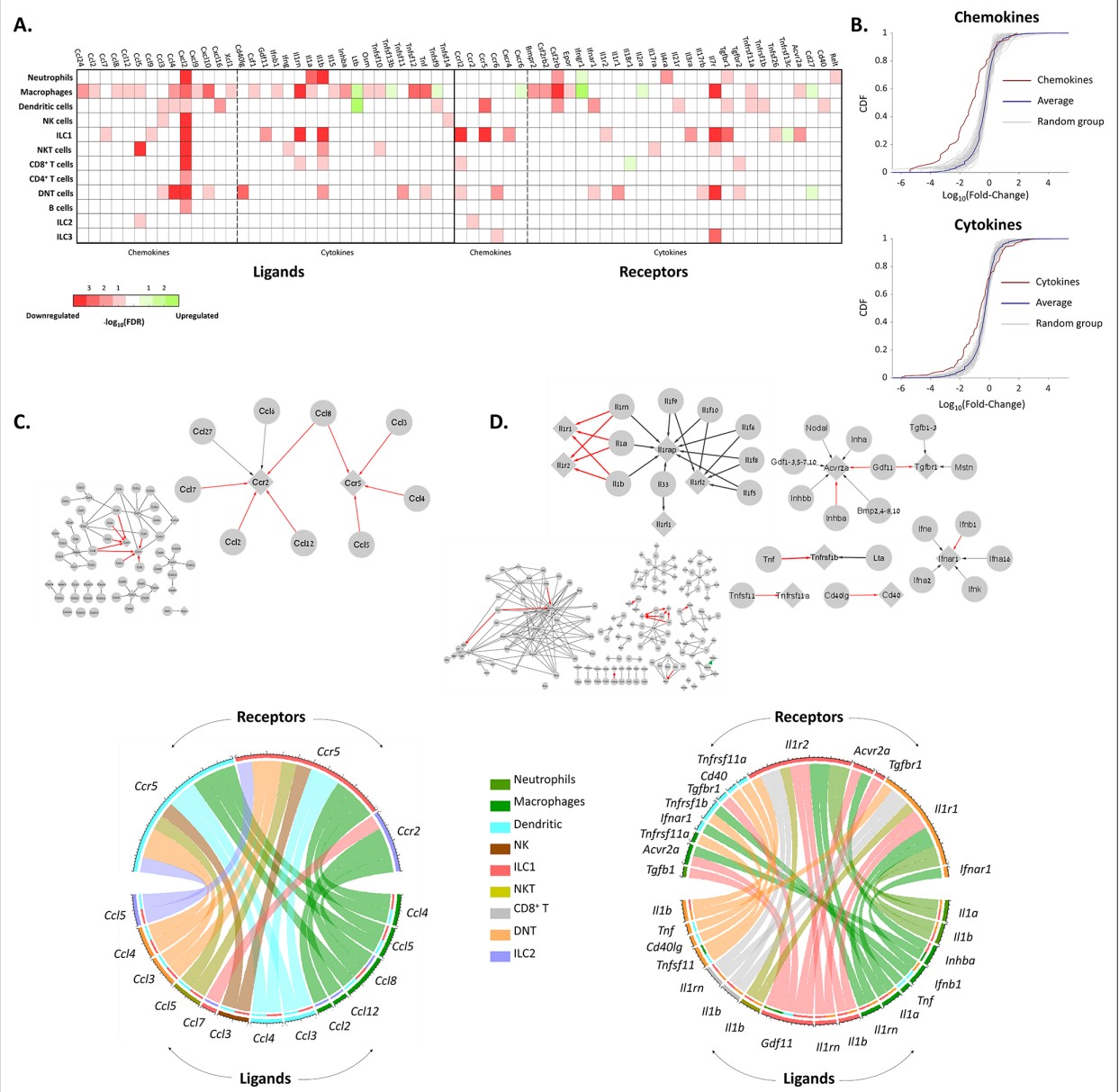

**Figure 4.** The effect of female age on the chemokines and cytokines networks of the ovarian immune cells. (**A**) A heat map of a significant (p-value <0.05; FDR ≤0.1) twofold decrease (red) or increase (green) in the expression levels of chemokines, cytokines, and their receptors in different immune cell types. (**B**) Cumulative probability distribution (CDF) of the fold change (FC) of chemokines (upper panel) and cytokines (lower panel) (red line). The gray lines are the CDFs of FC for random groups of genes at the same size as the chemokines/cytokines genes. The blue line is the CDF of the FC of all the genes. There is a significant decrease in the expression of chemokines and cytokines with age (Kolmogorov-Smirnov test p-value <0.01). (**C**) Downregulation of the chemokines network due to age. Upper panel - Edges within the chemokine network in which both the ligand and the receptor were significantly downregulated in at least one cell type are colored in red. Edges that only the ligand/receptor, or none of them, were significantly downregulated are colored in grey. The sub-graph that contains the affected interactions is magnified at the right-hand side of the figure. Bottom panel - Chord diagram that illustrate the decrease in chemokine ligand-receptor interactions between the different cell types. The color of each chord denotes the color of the cell type that underwent a reduction in ligand expression. In the upper semi-circles, the colors indicate the cell type that showed a decrease in receptor expression. In the lower semi-circle, the outer and inner colors denote the cell types in which ligands and target receptors were downregulated, respectively. (**D**) Downregulation in the cytokines network due to age. Same color-coding as in (**C**) for the cytokines interaction network.

The online version of this article includes the following figure supplement(s) for figure 4:

**Figure supplement 1.** Age differences in chemokines network are mainly due alternations in CCR5 expression.

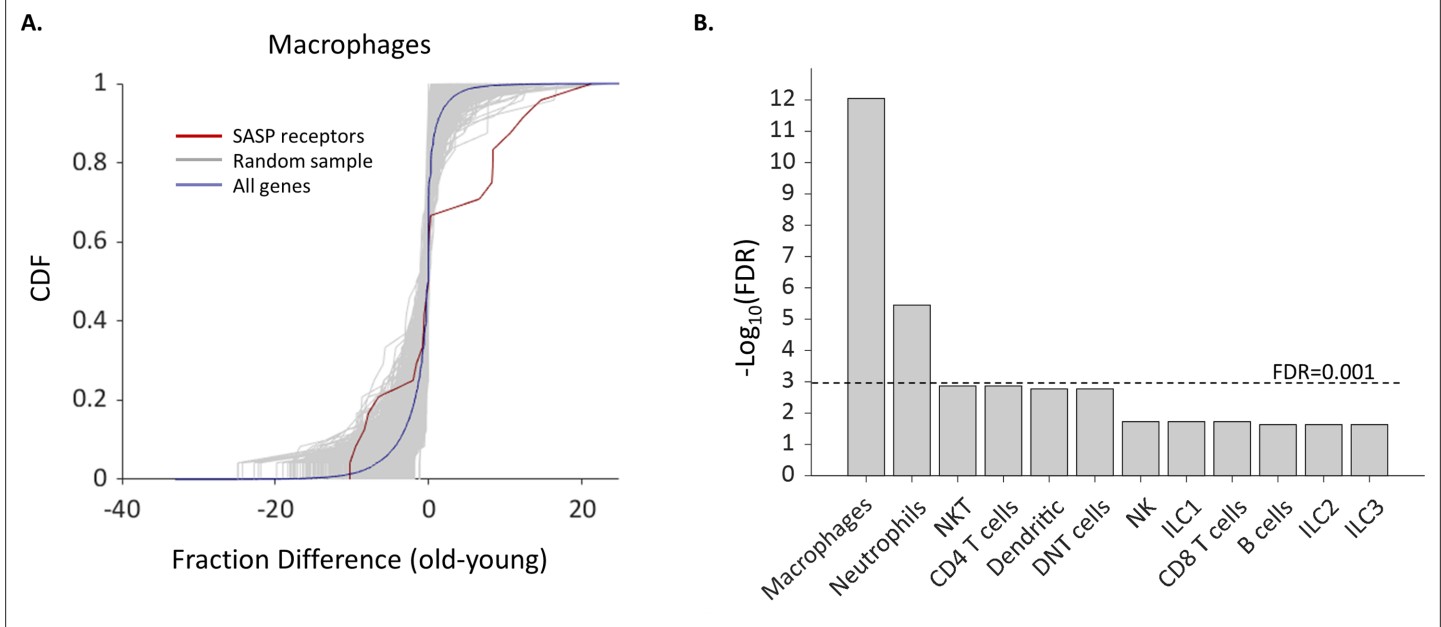

**Figure 5.** The fraction of Macrophages and Neutrophils expressing SASP receptors is elevated in old age. (**A**) Cumulative probability distribution (CDF) of the difference of a fraction of old and young macrophages that express 24 SASP genes (red line). The gray lines are the fraction difference CDFs of groups of 24 random genes (10,000 samples). The blue line is the CDF of the fraction difference of all the macrophage genes. There is a significant increase with age in the fraction of cells that express members of the SASP genes (Kolmogorov-Smirnov test p-value <0.01). (**B**) The false discovery rate (FDR) for significant change (p-value <0.01) in the fraction of different cell types that express SASP genes. Macrophages and neutrophils exhibit FDR which is much smaller than 0.001.

The online version of this article includes the following figure supplement(s) for figure 5:

**Figure supplement 1.** Cumulative probability distribution (CDF) of the fold change between old and young macrophages' expression in SASP genes.

and may result in chronic diseases and tissue dysfunction (***Muñoz-Espín and Serrano, 2014***; ***Ovadya and Krizhanovsky, 2014***). One of the main molecular features of senescent cells is the senescence-associated secretory phenotype (SASP), in which the senescent cells create an inflammatory environment by secreting inflammatory cytokines, chemokines, growth factors, extracellular remodeling factors, and more (***Prata et al., 2018***; ***Song et al., 2020***). Immune cells respond to these factors, detect specific markers expression or their absence on the senescent cells' membrane and clear them either by phagocytosis (by Mφs, for example) or by killing (by NTs or NK cells for example) (***Song et al., 2020***). Ovarian senescence was already studied in the past (***Velarde and Menon, 2016***); however, the specific mechanism of senescent cells clearance within the ovaries is still unclear.

We compiled a list of SASP receptors based on the literature containing 24 receptors (***Supplementary file 5***) and examined how their expression in different cell types depends on the female age. We found that the fraction of cells that express *Ccr2*, *Csf2ra*, and *Csf1r*, which are all receptors for known SASP proteins (***Rhinn et al., 2019***; ***Song et al., 2020***), was significantly higher in old Mφs. In addition, the fraction of old Mφs that express cell surface markers that were previously reported to take part in the recognition of senescent cells, such as membrane IgM's (*Ighm*) and C-type lectin receptors (*Clec4a2-3*) (***Burton and Stolzing, 2018***) was also elevated. Moreover, old NTs and Mφs showed upregulated expression of *Ifngr1* (***Figure 4A***), a part of the IFNγ receptor, while the cytokine itself is overexpressed by senescent cells (***Lujambio et al., 2013***; ***Pan et al., 2021***). In addition, both NTs and Mφs, as well as NK cells showed a higher expression fraction of this receptor. In total, we found six SASP receptors that were significantly overexpressed by old Mφs. The probability that six genes would be significantly modulated (p<0.01) out of a list of 24 random genes is low (FDR = $10^{-12}$, ***Figure 5***). Moreover, across all cell types, only old Mφs and NTs presented a significant elevated fractions of cells expressing SASP receptors (***Figure 5B***). As a complementary analysis, we checked the expression levels of all SASP receptors. Results show that SASP receptors expression in ovarian Mφs is not altered at old age (***Figure 5—figure supplement 1***). NTs also exhibited elevated fractions of *Ccr1*, a receptor for several SASP chemokines (***Coppé et al., 2010***), while old NKT cells had higher levels

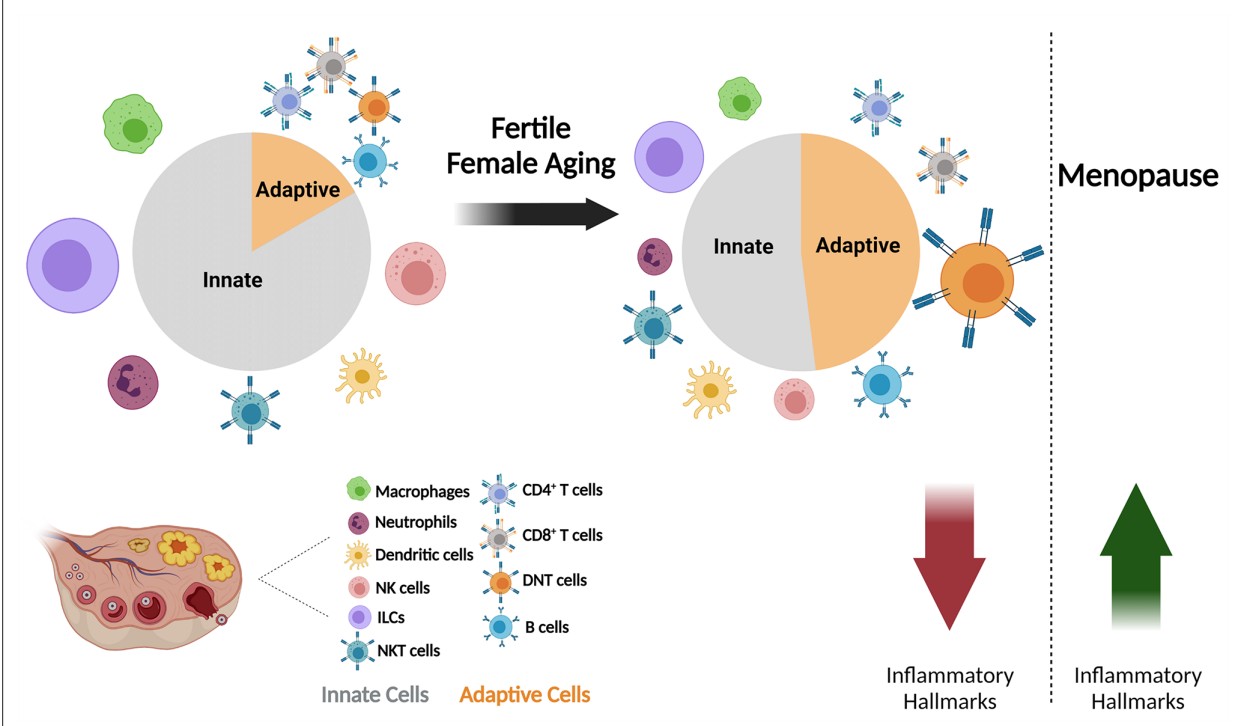

**Figure 6.** The effect of fertile female aging on the ovarian immune system. As the female ages, most innate immunity cells, such as NTs, Mφs, ILCs, and NK cells exhibit a decrease in their fractions within the total immune population. Moreover, there is a substantial increase in the fraction of DNTs. The ovarian immune aging during the female fertile period is characterized by decreased inflammatory hallmarks, as opposed to the post-menopause inflammaging. (Created with BioRender.com).

of *Cd74*, a receptor for MIF, another SASP member (*Coppé et al., 2010*; *Kim et al., 2018*). CXCR6 is another novel mediator of senescence control which was recently discovered as part of senescence surveillance in the liver by CD4 T and NKT cells (*Mossanen et al., 2019*). *Cxcr6* expression was also increased in old Mφs.

## Discussion

In this work, we characterized the first complete mouse ovarian immune milieu and its changes with female age up to near-estropause state, and discovered major changes in the fractions of different immune cell types (*Figure 6*). Our results present an elevation in CD3+ TCRβ+ lymphocytes along with age, indicating a dramatic change in the fraction of DNT cells (from ~5% to ~35% at old age). Other identified DNT cells, in the blood or lymph nodes, for example, consist of 1–5% out of all the lymphocytes (*Hillhouse and Lesage, 2013*; *Juvet and Zhang, 2012*). These results are consistent with a previous study that showed an increase in the TCRβ+ lymphocytes that is not due to CD4 or CD8 T cells change in mice before estropause (*Lliberos et al., 2021*). Double-negative T cells were found to be the most common lymphocyte across the female mice uterus and cervix (*Johansson and Lycke, 2003*). These cells were shown to have a regulatory function, showed no proliferation, inhibited the proliferation of splenic T cells when co-cultured together in vitro, and their origin was suggested to be extrathymic. The DNT population we observed in the ovary, showed almost no expression of classical thymocytes development markers at the double-negative stages such as Notch1, Socs3, Dtx1, Hes1, and more (*Puthier et al., 2004*). Other studies showed that DNT cells in peripheral blood or lymphoid organs have a suppressive role and assist in preventing allograft rejection and autoimmune responses (*Hillhouse et al., 2013*; *Juvet and Zhang, 2012*). Some studies suggest that DNT cells are the result of down-regulation of CD4 and CD8 due to chronic stimulation (*Rodríguez-Rodríguez et al., 2015*; *Grishkan et al., 2013*). The ovaries present ongoing cycles of inflammation processes in each ovulation, which may result in chronic stimulation of ovarian T cells and may be consistent with an increase of CD4 T cells after estropause (*Lliberos et al., 2021*). Furthermore, enrichment in CD3+ lymphocytes

in the ovary was found to be associated with poor follicular reserve (*Jasti et al., 2012*) and thus may have an impact on ovarian autoimmune diseases. Moreover, one of the characteristics of polycystic ovary syndrome (PCOS) is an abnormal T lymphocyte milieu that is suspected to be involved in disease pathogenesis and ovarian dysfunction that leads to infertility (*Li et al., 2019*). Similarly, our results indicate an age-dependent change in the T lymphocytes population that may be involved in improper immune regulation in the ovary and hence lead to infertility. In addition, ILC1 cells are the largest population in the young mouse ovaries, and the second largest in the old ovaries, after the DNT population. These cells, along with NK cells, constitute the group 1 innate lymphoid cells. As opposed to NK cells, ILC1 have weak-to-no cytotoxic capabilities, and are more similar to Th1 cells in their function, that is, inducing type 1 immune response against intracellular pathogens and inflammatory responses (*Vivier et al., 2018*). In addition, ILC1 cells take part in the process of tissue remodelling (*Jowett et al., 2020*). Interestingly, ILC1 distribution was also found to be elevated in the epididymal adipose of male mouse testis (*Boulenouar et al., 2017*). As a dynamic, constantly changing environment, the ovaries, which involve cycles of developing and regressing structures, may use ILC1 cells as orchestrators of other cell types participating in these processes. Our results also show a decline in the macrophages population with age, which may result in fertility deterioration because of impaired ovarian vasculature (*Turner et al., 2011*).

Taking together the observed changes in global processes and expression of immune mediators such as chemokines and cytokines, a state of general attenuation and unresponsiveness is emerging. GO enrichment analysis of joint processes for several cell types present a decrease in general activity, in which immune cells exhibit lower levels of metabolism-related processes accompanied by decreased responsiveness to different stimuli and loss of regulation over various processes. Analysis of the chemokines and cytokines networks also revealed general attenuation and decreased expression of various inflammatory mediators. Chemoattractants, such as *Cxcl2*, *Ccl2*, *Ccl3*, *Ccl4*, and *Ccl5*, as well as pronounced inflammatory cytokine transcripts such as TNFα, IFNγ, IL1-α, and IL-1β were all decreased in different immune cell types. Moreover, decreased expression of several members of the TGFβ superfamily by macrophages at old age, such as *Inhba* and *Acvr2a*, might be connected to impaired or decreased follicles growth. While previous studies suggested an inflammatory state in the post-estropausal ovaries (*Lliberos et al., 2021*), our results indicate a decrease in the inflammatory characteristics of ovarian immune cells during pre-estropause aging. We have found an increase in genes involved in senescent cell recognition by several immune cell types, mostly Mφs but also NTs, NK, and NKT cells. Generally, as age progresses, senescent cells are accumulating within tissues (*Childs et al., 2015*; *van Deursen, 2014*; *Velarde and Menon, 2016*). In the ovaries, aging and senescence are accompanied by an increasing number of atretic follicles and cycle irregularities that end in estropause. Our results suggest that clearance of senescent ovarian cells by immune cells increases at old age to match the ovary needs.

There is a plethora of studies that suggest inflammaging - an age-related increase in the inflammatory environment that develops over time (*Ferrucci and Fabbri, 2018*; *Franceschi et al., 2018*; *Giunta, 2006*). Previous recently published papers who characterized the age effect in the single-cell level on the immune system in various fertility non-related tissues have demonstrated inflammaging (*Almanzar et al., 2020*; *Kimmel et al., 2019*; *Mogilenko et al., 2021*). There are several major differences between these papers to our research. First, these papers compared not just females, but also males. Moreover, they used very old, post-estropausal mice and analyzed mostly tissues that are not part of the reproductive system. Our results show that the most evident change in the ovaries of aging female mice before estropause is a shift from innate to adaptive immunity, which is associated with a dramatic increase of CD3$^+$ lymphocytes, and a decrease in Mφs, NTs, and ILC1 fractions. Besides the changes in the composition of the immune milieu, most cell types exhibit a decrease in inflammatory hallmarks. Our results portray a novel intermediate state, in which prior to the elevation in inflammatory hallmarks and the development of inflammaging, there is an apparent decrease in inflammatory characteristics in the ovary (*Figure 6*). Our results suggest that ovarian immune aging is not linear, but a complicated process that exhibits alternations between anti- and pro-inflammatory environment.

## Methods

### Mice

All experiments involving mice conform to the relevant regulatory standards (Technion IACUC and national animal welfare laws, guidelines, and policies). Hsd:ICR female mice were purchased from Envigo RMS (Israel) and were housed in a 12 hr light / 12 hr dark cycle. Assessment of estrous cycle stage for the mice begun 3 days prior each experiment via vaginal smears (*McLean et al., 2012*). Briefly, mice's vaginal canals were washed using 20 µL saline (PBS, 0.09% NaCl). The saline was then collected and mounted on a slide and observed under an inverted microscope (Olympus) equipped with X20 objective. The estrous cycle-stage was assessed according to epithelial cells morphology and the presence or absence of leukocytes. Mice that exhibited regular progress of their cycle for three consecutive days were eligible for further experiments.

### Ovaries extraction and handling

Mice were anesthetized for 2 min using isoflurane and then euthanized by cervical dislocation. Each mouse's ovaries were extracted and washed in RPMI-1640 media (Sigma-Aldrich) containing 10% FBS and transferred to 1.5 mL microcentrifuge tubes containing the same media. Next, the ovaries were cut using scissors and incubated for 30 min with ~7800 IU/mL (Lot dependent) of collagenase type IV (Sigma-Aldrich) at 37°C. After which, 2.5µ g/mL of DNase I (Sigma-Aldrich) was added, and tubes were incubated for additional 30 min. For proper tube content mixing, gentle tapping was performed every 5 min during incubation. Tissue homogenate was filtered through a 40 µm strainer (Greiner Bio-One) and washed with 0.5 mL RPMI media five times. Total cell count was then calculated using LUNA automated cell counter (Logos Biosystems). Cells were further processed for sorting (for single-cell RNA sequencing) or staining (for flow cytometry experiments).

### Cell sorting

Cell samples went through a series of centrifugations (400$g$, 5 min, at 4°C) and were stained with PE anti-CD45 (30-F11, BioLegend) in staining buffer (see flow cytometry section) for 30 min at 4°C. CD45-positive cells were sorted and collected using FACSAria III Cell Sorter (BD Biosciences). Collected samples were centrifuged and brought to a final volume of ~50 µL, counted using LUNA automated cell counter (Logos Biosystems), and further processed for single-cell sample preparation.

### Single-cell RNA sequencing

Samples were prepared as outlined by the 10x Genomics Single Cell 3' v2 Reagent Kit user guide. Briefly, the samples were washed twice in PBS (Sigma-Aldrich) +0.04% BSA (Sigma-Aldrich) and re-suspended in PBS + 0.04% BSA. Sample viability was assessed with Trypan Blue (Biological Industries) using LUNA automated cell counter (Logos Biosystems) and validated using a hemocytometer. Following counting, the appropriate volume for each sample was calculated for a target capture of 10,000 and 7700 cells (old and young samples, respectively) and loaded onto the 10x Genomics single-cell-A chip. After droplet generation, samples were transferred onto a 96-well plate, and reverse transcription was performed using a Veriti 96-well thermal cycler (Thermo Fisher). After the reverse transcription, cDNA was recovered using Recovery Agent provided by 10x Genomics followed by a Silane DynaBead clean-up (Thermo Fisher) as outlined in the user guide. Purified cDNA was amplified before being cleaned up using SPRIselect beads (Beckman Coulter). Samples were quantified with Qubit Fluorometer (Invitrogen) and run on Agilent TapeStation for quality control. cDNA libraries were prepared as outlined by the Single Cell 3' Reagent Kits v2 user guide with appropriate modifications to the PCR cycles based on the calculated cDNA concentration (as recommended by 10x Genomics). Post library construction quantification and QC were performed as mentioned above for the post cDNA amplification step. Sequencing was performed with NextSeq500 system (Illumina), with ~50,000 reads per cell and 75 cycles for each read.

Seurat R package (*Butler et al., 2018*) was used to read the 10x output data from Cell Ranger v3.0.1 according to the suggested protocol. Yield was 3693 and 5644 cells for young and old samples, respectively. Quality control tests were conducted to eliminate duplicates, and dead or low-quality cells – cells with less than 200 features, more than 2500 features, or more than 10% features of mitochondrial genes were excluded from further analysis. In total, we ended up with 3307 and 5468 cells for young and old samples (data available at *Figure 1—source data 1* and *Figure 2—source data 1*).

## Cell type annotations

The Seurat R package (version 4.1.0), along with the SingleR package (version 1.8.1) (*Aran et al., 2019*) were used to analyze single-cell RNA sequencing data. After using Seurat's normalization tool ('LogNormalize', 10,000 scale) and clusters identification using Seurat's graph-based clustering method, t-distributed stochastic neighbor embedding (tSNE) was used as a dimension reduction and visualization tool, with PCA as the latent space (15 PC dimensions). The log-normalized data was scaled prior to dimensionality reduction (using 'scale.data'). Next, the SingleR algorithm was used to achieve the automatic annotation for each cell. Briefly, the algorithm compares each cell's transcriptome to known transcriptomic 'signatures' from reference genomes taken from The Immunological Genome Project (ImmGen) (*Heng et al., 2008*). The algorithm calculates the correlation between the cell to different cell types, and based on the highest correlation suggests an annotation for the cell.

In addition, a set of literature-based gene markers was chosen to identify the different immune cell types. For each gene marker, a normalized score was calculated in every cell using the raw count matrix. $E_{g,i}$ denotes the expression level of the gene $g$ within the $i^{th}$ cell; $M_g$ and $m_g$ denote the maximal and minimal expression of the gene $g$ across all cells in the sample, respectively. $N_c$ denotes the number of cells within cluster $c$, and $N_{g,c}$ denotes the number of cells within cluster $c$ that express the gene $g$. The normalized expression score, $S_{g,i,c}$, for the gene $g$ in the $i^{th}$ cell within the cluster $c$ is the multiplication of the gene relative expression and the relative fraction of cells within the cluster that express this gene, $S_{g,i,c} = \left( \frac{E_{g,i} - m_g}{M_g - m_g} \right) \cdot \left( \frac{N_{g,c}}{N_c} \right)$. *Figure 1c* illustrates the distribution of the normalization score for a particular gene $g$ over all the cells in cluster $c$. Normalized scores are between 0 and 1.

The identity of each cluster was determined by taking into account the majority type as given by the SingleR and the type according to the literature markers. For example, according to the literature markers, cluster 10 is $CD8^+$ enriched, and cluster 11 is $CD4^+$ enriched. While SingleR cell type ID is consistent with this classification, several cells in cluster 10 were classified as $CD4^+$ cells, while other cells in cluster 11 were classified as $CD8^+$ cells. Therefore, the final assignment of these cells is different than the SingleR ID.

## Flow cytometry analysis

Ovarian cell suspensions were stained for flow cytometry analysis as the commercial protocol suggested (BD Biosciences). Briefly, cells were plated in 96-well U-shaped plates and went through a series of centrifugations ($400g$ for 5 min, at 4°C) for media and debris cleaning. Next, samples were stained as the commercial protocol suggested using the following antibodies (purchased from BioLegend) diluted in staining buffer (PBS containing 0.09% sodium azide (Sigma-Aldrich)): PE anti-CD45 (30-F11), BV421 anti-CD11b (M1/70), APC/Cy7 anti-F4/80 (BM8), APC anti-F4/80 (BM8), BV711 anti-CD11c (N418), Alexa Fluor 700 anti-Ly6G (1A8), APC anti-CD3 (17A2), Pacific Blue anti-CD3 (145–2 C11), APC/Cy7 anti-NK1.1 (PK136), PE/Cy7 anti-TCRβ (H57-597), Alexa Fluor 700 anti-CD8a (53–6.7), FITC anti-CD4 (GK1.5), PE/Cy7 anti-TCRγδ (GL3) and APC anti-CCR5 (HM-CCR5). Staining was performed at 4°C for 30 min. Flow cytometry analyses were made using the S100EXi (Stratedigm) flow cytometer. Viability test of $CD45^+$ cells was conducted using Zombie-NIR (BioLegend).

## Statistical analyses

All statistical analyses were calculated using MATLAB R2019b (MathWorks).

For the flow cytometry experiments, we used two-sample Kolmogorov-Smirnov test for identifying significant changes in cell types fractions at different ages.

To determine DEGs between old and young samples, genes were considered significant if they had twofold change in their mean expression and if their FDR ≤0.1, using Storey q-values approach (*Storey, 2002*) for a p-value ≤0.05 (two-tailed student's t-test).

For analyzing significant changes in the fraction of each cell type, we used the MILO algorithm (*Dann et al., 2022*). Briefly, the algorithm constructs an undirected KNN graph of single cells based on the scRNA-seq, divides the data into neighborhoods of cells and compares how their size changes with different conditions. Therefore, we have used the batch-corrected data as input for it. To estimate the differential abundance in terms of cell types, the MILO algorithm assigns a cell type label to each neighborhood by finding the most abundant cell type within cells in each neighborhood (*Figure 2—figure supplement 1*). MILO uses a weighted version of the Benjamini-Hochberg method to regulate

the spatial FDR in the KNN graph. In this method, p-values were weighted by the reciprocal of the neighborhood connectivity to regulate the graph.

The FDR of having six genes that have significantly modulated fraction change (p-value <0.01) out of a list of 24 random genes, for each cell type, was calculated analytically. The FDR in this case, where all hypotheses are considered null, is equal to the family-wise error rate (FWER), which can be calculated analytically using multinomial distribution (*Korthauer et al., 2019*).

### GO enrichment analysis

Significant downregulated or upregulated genes for each cell type were taken for GO enrichment analysis of biological processes using the g:Profiler platform (*Raudvere et al., 2019*) (database versions: Ensembl 108, Ensembl Genomes 55, and Wormbase ParaSite 17, published on 2/23/2023). Next, all significant processes were further analyzed using the REVIGO platform for eliminating redundant GO terms (*Supek et al., 2011*). Then, cell-type specific and global (for at least two cell types) processes were found and analyzed once more using the REVIGO platform in order to cluster them according to their semantics distance.

### Cytokines and chemokines networks

The Kyoto Encyclopedia of Genes and Genomes (KEGG) was used to construct the chemokines and cytokines ligand-receptor networks. These were built based on the Cytokine-cytokine receptor interaction - *Mus musculus* (mouse) map (pathway map mmu04060) (*Kanehisa, 2000*).

## Acknowledgements

We thank Nati Karin, Keren Yitzhak, Noam Kaplan, and the Savir lab members for fruitful discussions. This work was supported by the Israel Science Foundation (ISF) grant #1619/20, the Rappaport Family Institute for Research in the Medical Sciences Thematic Grant, and the Wolfson Foundation.

## Additional information

### Funding

| Funder | Grant reference number | Author |
| --- | --- | --- |
| Israel Science Foundation | 1619/20 | Tal Ben Yaakov<br>Tanya Wasserman<br>Eliel Aknin<br>Yonatan Savir |
| Rappaport Family Institute for Research in the Medical Sciences | | Tal Ben Yaakov<br>Eliel Aknin<br>Yonatan Savir |
| Wolfson Foundation | | Yonatan Savir |

The funders had no role in study design, data collection and interpretation, or the decision to submit the work for publication.

### Author contributions

Tal Ben Yaakov, Conceptualization, Data curation, Software, Formal analysis, Validation, Investigation, Visualization, Methodology, Writing – original draft, Writing – review and editing; Tanya Wasserman, Conceptualization, Data curation, Validation, Investigation, Visualization, Methodology, Writing – review and editing; Eliel Aknin, Software, Validation, Visualization, Methodology, Writing – review and editing; Yonatan Savir, Conceptualization, Resources, Data curation, Software, Formal analysis, Supervision, Funding acquisition, Validation, Investigation, Visualization, Methodology, Writing – original draft, Writing – review and editing

### Author ORCIDs

Tal Ben Yaakov (ORCID) http://orcid.org/0009-0005-0550-2901

Tanya Wasserman http://orcid.org/0000-0003-0221-1891
Yonatan Savir http://orcid.org/0000-0002-5345-8491

### Ethics

All mouse experiments performed in this study were approved by the Animal Care and Use Committee of the Technion, Israel Institute of Technology, and found to confirm with the regulations of this Institution for work with laboratory animals, protocol No: IL-069-05-2021.

### Decision letter and Author response

Decision letter https://doi.org/10.7554/eLife.74915.sa1
Author response https://doi.org/10.7554/eLife.74915.sa2

## Additional files

### Supplementary files

- Supplementary file 1. Up and down DEGs.
- Supplementary file 2. Summary of downregulated processes REVIGO.
- Supplementary file 3. Summary of upregulated processes REVIGO.
- Supplementary file 4. GO enrichment CDF based.
- Supplementary file 5. SASP receptors list.
- Transparent reporting form

### Data availability

All data used in this study are included in the manuscript, the supporting files and in GitHub:https://github.com/SavirLab/AgingOvarianImmuneMilieu (copy archived at *SavirLab, 2023*).

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
