## [Editor Report]

The study describes a single-cell analysis of the mammalian ovary in young, adult, and old mice, and is an important contribution to the field identifying clusters of immune cell populations across the different ages. The combination of single-cell RNA sequencing and flow cytometry used is a robust and unbiased approach that provides compelling evidence of immune cell alterations in aged ovaries.

---

## [Decision Letter]

**Decision letter after peer review:**

Thank you for submitting your article "Single-cell analysis of the aged ovarian immune system reveals a shift towards adaptive immunity and attenuated cell function" for consideration by *eLife*. Your article has been reviewed by 3 peer reviewers, and the evaluation has been overseen by a Reviewing Editor and Ricardo Azziz as the Senior Editor. The reviewers have opted to remain anonymous.

Essential revisions:

1. Figure 1-2 begins to tell an interesting story of changes in macrophage and an ovary specific CD3^+^ CD4^-^ CD8^-^ double negative T-cell abundances in aging. While the flow cytometry here backs up their claims, the statistical rigor of the single-cell analysis itself is questionable. Rather than further probing gene expression differences in these specific subpopulations, the next sections proceeds to do (1) a global analysis of gene expression changes, (2) a CCC analysis (independent of existing tools) that claims decreased inflammation (see below points about doubts regarding this), and (3) an analysis of SASP recognition that is very limited.

2. Results for Figure 1-3 can be published with some changes. Figure 4 would need a lot of additional analyses and changes. The SASP section with no main figure associated shouldn't be included.

3. Code for computational analyses should at least be available on Github, preferably on CodeOcean for reproducible runs.

4. My concern regarding the single-cell results is that there does not seem to be a formal batch correction step. Figure 2A visually seems to show minimal batch effects, and a big positive is that flow cytometry results align with the single-cell observations. Unfortunately, it is impossible to rigorously claim from the scRNAseq that the cluster 12 frequency change is due to aging without a formal batch correction.

5. The problem with applying a batch correction now, even if there aren't major batch effects, would be that it can change downstream results at some resolution (e.g. p-values and effect sizes). A solution would be to add a supplemental section demonstrating that there are not batch effects. This may be done by applying a batch correction (e.g., Harmony or Seurat integration), and demonstrating that downstream clustering patterns remain similar (indicating that the informative transcriptional space of the cells are consistent).

6. A separate but complementary point regarding the question of frequency change in cluster 12 using scRNAseq: unlike microbiome, which has comprehensive compositional analysis methods, the question of cell abundance changes in single-cell is just recently beginning to be addressed. To more rigorously present these results, a few things would be useful:

6.1 Higher resolution in Figure 2C,D – specifically, gating on the CD3^+^ lymphocyte subpopulations, as well as some control cell types that do not show a change in single-cell (preferably, all cell type frequencies validated with flow).

6.2 More quantification of effect size or statistical assessment of Figure 1B using recently published tools. Some tools have been published on differential abundance testing in single-cell in the last couple years include scDC, MILO, and DA-seq.

7. As a side note, MELD does not give a differential abundance p-value, but does quantify the likelihood of observing a given cell in a given condition at single-cell resolution and allows you to further partition the data based on those values. This can allow for higher resolution differential expression testing and may be useful to you for future analyses.

8. There is a lack of multiple test correction (or stating of such correction) throughout statistical analyses which must be addressed.

9. It is difficult to trust the "skewed" DE patterns, especially for DNT cells (Figure 3A) -- a global downregulation of genes?

10. The claim regarding a decreased inflammatory state in aging is unconvincing. Currently, the results indicate a global downregulation of the transcriptome in aging, and so when you visualize just chemokines and cytokines, visually, it looks like inflammation is downregulated. The enrichment analysis would be better in supporting this claim -- the story could go: inflammatory response is a consistently enriched term in cell types X, Y, Z [the Results section regarding Figure 3], so we then focused on communicatory immune networks [Results section regarding Figure 4]. I had formatting issues in Supplementary Table 2, but it looks like the inflammatory response GO Term is only enriched in macrophages. Further discussion should be had to back up this claim.

11. Senescent cells section seems like an afterthought. It is not sufficient to make the claim of increased senescent cell recognition by immune cells via single-cell analysis of immune cells alone, and even if it was, these analyses are not rigorous enough.

12. There is no visualization of canonical SASP receptors expression changes across young vs old.

13. Analysis is not systematic: should start with a comprehensive list of canonical SASP receptors, rather than choosing some from literature.

14. Supplementary Figure S5 is an unorthodox analysis of gene expression changes. Were these genes differentially expressed in old macrophages?

15. Throughout the paper, it is important to show (by either experiments or using publicly available resources) that the changes observed are indeed specific to the ovaries.

16. The authors start the paper by discussing the reduced fertility as a function of age, so any of these results suggest a mechanisms for that? Some discussion of this point will be useful.

17. Because the scRNA-seq data presented by the authors show that the CD4^-^ CD8^-^ double-negative T cell subset co-express Trbc2 (TCRb) and Tcrgc2 (TCRg) genes, it would be important to test if these cells also co-express TCRb and TCRg/d at the protein levels. Pro-inflammatory CD4^-^ CD8^-^ double-negative T cells co-expressing TCRb and TCRg/d have been found in mice (Edwards et al., J Ex Med 2020), and it would be interesting to test whether the ovarian DNT cells show phenotypical or functional similarities with this cell type.

18. To better understand the function of double-negative T cell subset in aging ovaries, one possible way would be to purify these cells and measure which cytokines they produce after TCR activation in vitro and/or co-culture these cells with activated CD4/CD8 T cells in vitro to test if they are capable of suppressing T cell proliferation.

19. For the cluster annotation of scRNA-seq data, it would be interesting to perform additional gene expression analyses to test whether the two clusters of dendritic cells correspond to cDC1 and cDC2 populations.

20. Flow cytometry validation of scRNA-seq data in larger groups of mice presented in this study is chiefly limited by CD3^+^ T cells and CD11b^+^ cells. Additional flow cytometry experiments that validate alterations of central ovarian immune cell populations in old competed to adult mice would be helpful. Gating strategies for all flow cytometry experiments should be shown.

21. It would be interesting to compare the scRNA-seq data generated by the authors with published datasets on the immune aging in various mouse tissue (e.g., Almanzar et al., Nature 2019; Kimmel et al., Genome Res 2019; Mogilenko et al., Immunity 2021) to identify common and tissue-specific immune changes in aging ovaries.

22. Predicted changes in cytokine and chemokine expression levels and the crosstalk between immune and senescent cells presented in this study are based on scRNA-seq data but are lacking additional validation. For example, protein-level confirmation for some of these pathways would add important information about the mechanism of immune aging in the ovaries.

23. In Methods: antibody clone 17A2 is used for CD3 and CD4 detection (possible mistake).

[Editors' note: further revisions were suggested prior to acceptance, as described below.]

Thank you for resubmitting your work entitled "Single-cell analysis of the aged ovarian immune system reveals a shift towards adaptive immunity and attenuated cell function" for further consideration by *eLife*. Your revised article has been evaluated by Ricardo Azziz (Senior Editor) and a Reviewing Editor.

The manuscript has been improved but there are some remaining issues that need to be addressed, as outlined below:

*Reviewer #1 (Recommendations for the authors):*

In this study, the authors present an exciting work that lies at the intersection of immunology, aging, and single-cell RNA-sequencing analysis. It provides a valuable and well-annotated single-cell resource for other researchers in the community to use. It provides solid analyses to probe mechanistic changes in ovarian immune cell functions. More specifically, it leverages single-cell RNA sequencing to probe changes in various immune functions within the ovary in aging. The data provided is the most comprehensive of ovarian immune cells at the resolution of single-cell transcriptomics to date and will be valuable to other researchers. The authors explore four distinct immune functions:

1. Among other cell types, the authors identify macrophages and a unique CD3^+^ CD8^-^ CD4^-^ T-cell (DNT) subpopulation that changes in abundance with aging. The cell-type compositional analysis results are comprehensive and convincing.

2. The authors also analyze changes in global gene expression across cell types using an enrichment analysis; Figure 3B summarizes potential global and cell-type specific changes in gene expression programs during aging.

3. The authors infer differences in cell-cell communication mediated by various chemokines and cytokines, reasonably demonstrating a decreased inflammatory response.

4. The authors provide evidence that the fraction of macrophages and neutrophils recognizing secretory-associated senescence phenotype (SASP) molecules increases with age.

Both the data and biology presented are quite interesting. The distinction between an aging-associated decreased inflammatory response and cytokine/chemokine communication and an increase in SASP recognition in some cell types, particularly macrophages, demonstrates the complexities of the immune response that are amenable to further exploration. The role of the unique DNT population, which demonstrates substantial compositional changes with aging, in these systemic changes will also be interesting to further dissect.

Overall, while the authors have extensively addressed most of my concerns regarding the compositional analysis, the claims around a decreased inflammatory state with aging (particularly Figure 4B and the response regarding GO terms including both positive and negative regulators), and cell-cell communication analysis. I find the distinction between Figure 5 and Figure 5—figure supplement 1 to be interesting, with SASP recognition seemingly affecting a larger fraction of macrophages but not the average expression between the conditions. I also find it interesting that this same cell type decreases in abundance with age, possibly indicating that a subpopulation of macrophages that are retained with age are those exhibiting SASP recognition as an alternative explanation to the more natural conclusion that macrophages overall increase SASP recognition over time. While I am excited about this work, there are still some outstanding concerns, that I present below.

– Were the log-normalized data scaled prior to dimensionality reduction? PCA typically takes scaled data as input.

– Reading through the methods, it is unclear whether the p-values used in DE testing were multiple tests corrected. Line 364 of the Related Manuscript File "100652_1_related_ms_2716393_rmry4f.pdf" states "all other" statistical analyses applied an FDR correction, implying that this wasn't applied to the DE and other statistical tests discussed in the "Statistical analyses" subsection of the Methods. Furthermore, in the Supplementary Table reporting DE results, there is only the column "pVal" indicating that this is not a multiple test corrected significance value. If multiple test correction was not applied to differential expression output p-values, they must be. Similar concerns for the GO enrichment results and MILO results, which also include many tests. Furthermore, given the LFC threshold filter, I wouldn't expect results to change drastically. However, if there are similar concerns to my original comments regarding how batch correction will affect downstream effect sizes, a demonstration that applying the multiple test correction does not change the results is a necessary minimum. I would suggest demonstrating that the genes identified as significantly differentially expressed with multiple test correction (perhaps with FDR ≤ 0.1) are consistent with those in the current list. This could be done by showing that the gene list sizes are similar and have a high Jaccard index.

*Reviewer #3 (Recommendations for the authors):*

The authors made substantial improvements to the manuscript by cross-referencing the data and adding validation experiments. However, limitations of this revised manuscript still include not optimal validation strategies: e.g., in flow cytometry validation experiments, the authors defined dendritic cells as CD45^+^ CD11c^+^ subset, which might include a variety of CD11c^+^ macrophages; ILC1s were defined as CD45^+^ NK1.1^+^ cells, which might consist of NKT cells. These limitations prevent direct comparison of scRNA-seq data with the results of biological validation experiments.

---

## [Author Response]

Essential revisions:1. Figure 1-2 begins to tell an interesting story of changes in macrophage and an ovary specific CD3^+^ CD4^-^ CD8^-^ double negative T-cell abundances in aging. While the flow cytometry here backs up their claims, the statistical rigor of the single-cell analysis itself is questionable. Rather than further probing gene expression differences in these specific subpopulations, the next sections proceeds to do (1) a global analysis of gene expression changes, (2) a CCC analysis (independent of existing tools) that claims decreased inflammation (see below points about doubts regarding this), and (3) an analysis of SASP recognition that is very limited.

We thank the reviewers for their constructive remarks. In the following, we provide a point-to-point response to their comments. The revision includes performing additional experiments to validate the single-cell RNA-seq results and our analysis and performing additional computational analysis that strengthens our conclusion.

2. Results for Figure 1-3 can be published with some changes. Figure 4 would need a lot of additional analyses and changes. The SASP section with no main figure associated shouldn't be included.

Following these important comments, we have performed additional analysis of the results presented in Figure 4 that includes demonstrating that the inferred interactions are indeed significant (see detailed description in the relevant points below 9-10). We added a figure that illustrates the increase of SASP in macrophages to the main text together with a detailed analysis of its significance (see detailed description in the relevant points below 11-14)

3. Code for computational analyses should at least be available on Github, preferably on CodeOcean for reproducible runs.

The code is available at GitHub at https://github.com/SavirLab/AgingOvarianImmuneMilieu

4. My concern regarding the single-cell results is that there does not seem to be a formal batch correction step. Figure 2A visually seems to show minimal batch effects, and a big positive is that flow cytometry results align with the single-cell observations. Unfortunately, it is impossible to rigorously claim from the scRNAseq that the cluster 12 frequency change is due to aging without a formal batch correction.5. The problem with applying a batch correction now, even if there aren't major batch effects, would be that it can change downstream results at some resolution (e.g. p-values and effect sizes). A solution would be to add a supplemental section demonstrating that there are not batch effects. This may be done by applying a batch correction (e.g., Harmony or Seurat integration), and demonstrating that downstream clustering patterns remain similar (indicating that the informative transcriptional space of the cells are consistent).

We thank the reviewer for these important comments (*4+5*). These comments raise two main questions that are related but different. The first is how one can determine the cell types within each condition. The second, given the cell types, how can one have some confidence in the estimation of the fraction of each cell type out of the population within each condition and confidence in the change in fraction between the two conditions. In our response to points 4,5 and 6, we address these two main issues.

Indeed, if inferring the cell types rely *solely* on clustering the two conditions together, batch correction is vital. In our case, we determined the cell types not only by using clustering the two conditions together but also curated the cell types by examining each condition *by itself*, using literature markers, and automated algorithms (SingleR) that infer the immune cell types for each condition by itself. That is, the tSNE plot that shows the overlay of the population aims to demonstrate that even without batch correction our identification of the cell types is consistent between the two conditions.

Following, the reviewer’s comment, we performed formal batch correction using Seurat and show that the dimension reduction and cluster distribution remain almost identical, and that our cell type identification holds. We added a figure that demonstrates that, to the supplementary information (Figure 1—figure supplement 2), and addressed it in the relevant parts of the revised manuscript.

There are two complementary approaches for having confidence in the estimation of the changes in cell fraction as a function of age. The first is using classical simultaneous confidence estimations for the cell type fractions for each condition by itself. The second is to perform differential analysis using tools such as MILO that accounts also for the cell’s space topology.

As the reviewer mentioned, when using the latter approach, it is crucial to use batch correction. When we performed this analysis, we used it on the batch-corrected data. (We elaborate on it in point 6.2 below).

6. A separate but complementary point regarding the question of frequency change in cluster 12 using scRNAseq: unlike microbiome, which has comprehensive compositional analysis methods, the question of cell abundance changes in single-cell is just recently beginning to be addressed. To more rigorously present these results, a few things would be useful:6.1 Higher resolution in Figure 2C,D – specifically, gating on the CD3^+^ lymphocyte subpopulations, as well as some control cell types that do not show a change in single-cell (preferably, all cell type frequencies validated with flow).

Following the reviewer’s comments, we performed additional flow cytometry experiments in which we characterized the CD3^+^ subpopulation at a finer resolution – we measured the change in NKT cells, CD4, CD8, and double negative T-cells. These results are shown in the revised Figure 2D and are consistent with the changes in cluster 12 as measured using the scRNAseq. In addition, we also validated the changes in the fractions of several innate immune populations such as dendritic cells, neutrophils, and group 1 ILC cells (ILC1 and NK cells). These results are shown in Figure 2—figure supplement 2 in the revised supplementary information.

6.2 More quantification of effect size or statistical assessment of Figure 1B using recently published tools. Some tools have been published on differential abundance testing in single-cell in the last couple years include scDC, MILO, and DA-seq.7. As a side note, MELD does not give a differential abundance p-value, but does quantify the likelihood of observing a given cell in a given condition at single-cell resolution and allows you to further partition the data based on those values. This can allow for higher resolution differential expression testing and may be useful to you for future analyses.

Following the reviewer’s remarks, we took a dual approach for estimating the confidence levels in the change of cell types’ abundance with age.

First, recall that Figure 2B illustrates the estimator for the fraction of each cell type, which is simply the ratio between the number of cells for a given type divided by the total number of cells, within each condition (which is the maximum likelihood estimator of a multinomial distribution). The simultaneous confidence level on this estimator can be calculated by using bootstrapping or by using approximated analytic expression. We used bootstrapping to calculate the 95% simultaneous confidence levels of the fraction and revised Figure 2B accordingly. This provides one type of confidence level in the changes between conditions.

Second, following the reviewer’s comment, we applied MILO to estimate the differentiation abundance between conditions. As we mentioned in our response to point 4, it is critical to use batch-corrected data (using Seurat in our case) because this method identifies local neighborhoods in the transcriptome space and evaluates the change in abundance locally. The results shown in Figure S9 in the SI. This analysis shows that indeed the change in cluster 12 is significant.

8. There is a lack of multiple test correction (or stating of such correction) throughout statistical analyses which must be addressed.

We thank the reviewer for this important comment. All the relevant statistical tests were adjusted to multiple tests. We mentioned it in the relevant places in the revised manuscript. The approach we took throughout the entire work was to use FDR.

9. It is difficult to trust the "skewed" DE patterns, especially for DNT cells (Figure 3A) -- a global downregulation of genes?10. The claim regarding a decreased inflammatory state in aging is unconvincing. Currently, the results indicate a global downregulation of the transcriptome in aging, and so when you visualize just chemokines and cytokines, visually, it looks like inflammation is downregulated. The enrichment analysis would be better in supporting this claim -- the story could go: inflammatory response is a consistently enriched term in cell types X, Y, Z [the Results section regarding Figure 3], so we then focused on communicatory immune networks [Results section regarding Figure 4]. I had formatting issues in Supplementary Table 2, but it looks like the inflammatory response GO Term is only enriched in macrophages. Further discussion should be had to back up this claim.

We thank the reviewer for these important comments (9+10). Following these comments, we carefully revised and clarified the argument regarding the decrease of the inflammatory state.

As the reviewer correctly mentioned, a general downregulation can lead to an artifact in which inflammation-related genes appear to be decreased although it is not significant under the global downregulation. To account for this global downregulation, we’ve performed another differentially expressed genes (DEGs) analysis for each cluster, considering only the top down/up regulated genes with a p-value of 0.025 (which were defined by CDF analysis). The GO enrichment analysis using the g:Profiler platform, “Inflammatory response” (GO:0006954) downregulated in four different clusters (Neutrophils, Macrophages, Dendritic cells and ILC3 cells). In addition, the process “negative regulation of neuroinflammatory response” (GO:0150079) was upregulated in NK cells (see Supplementary File 5).

Yet, it is important to consider that enrichment of the GO term “Inflammatory response” (GO:0006954) is not necessarily an indicator of an inflammatory state. This term includes genes that are positive regulators of inflammation as well as negative ones. That is, one cannot infer whether inflammation goes up or down just because the GO term went up or down. This is true for other GO terms that involve immune response processes.

Therefore, in order to determine whether inflammation-related processes are decreased, one has to carefully examine the mediators of immune response – chemokines and cytokines.

Following the reviewer’s comment, we added the analysis that shows that even on top of the global downregulation, chemokines and cytokines are significantly downregulated in the cells from old ovary. This analysis was added as Figure 4B in the revised manuscript. The question following this step is whether the significant decrease in chemokines and cytokines indeed translates to a decrease in inflammation.

To answer this question, one must consider both the identity of the significantly downregulated chemokines and cytokines and their interaction across different cell types.

It is important to note that we did not cherry-pick the interactions that appear in Figure 4, but included all known chemokines and cytokine interactions. Most of the significant chemokines and cytokines changes were inflammatory interactions that were downregulated. Following the reviewer’s comments, we revised Figure 4 so it now contains the entire chemokine and cytokine networks.

To validate our results further, we applied cell2cell (Armingol et al., 2022) to our data to infer the age-dependent interactions. Consistently with our analysis, it yielded CCR5 as the main receptor that is modulated by age (Figure 4—figure supplement 1A).

Finally, to experimentally validate our results, we measured the fraction of cells that express CCR5 – and showed that indeed it decreases significantly (Figure 4—figure supplement 1B).

11. Senescent cells section seems like an afterthought. It is not sufficient to make the claim of increased senescent cell recognition by immune cells via single-cell analysis of immune cells alone, and even if it was, these analyses are not rigorous enough.12. There is no visualization of canonical SASP receptors expression changes across young vs old.13. Analysis is not systematic: should start with a comprehensive list of canonical SASP receptors, rather than choosing some from literature.14. Supplementary Figure S5 is an unorthodox analysis of gene expression changes. Were these genes differentially expressed in old macrophages?

We thank the reviewers for these important comments (11-14) regarding the SASP analysis. Following these comments, we have thoroughly revised this section and added a new figure to the manuscript (Figure 5 in the revised manuscript and Figure 5—figure supplement 1 in the revised SI).

First, following the reviewers’ comments, we have compiled a curated list of SASP receptors. While there is a wide milieu of SASP-related ligands and receptors in the literature, there is no single canonical list of SASP receptors. After an extensive literature review, we have compiled a list that contains 24 SASP-related receptors based on the available databases (such as KEGG) and literature. (Supplementary File 4).

Besides analyzing the change in gene expression (for example, by compiling the volcano plots that appear in Figure 3A in the revised manuscript) it is also crucial to consider the change in the fraction of cells that express a particular gene. In many immune-related processes, the fraction of cells that express a particular receptor determines the response. This is the reason that in many immune studies, the determination of whether an immune cell-type population was changed is based on the change in the fraction that expresses a particular receptor rather than expression fold change.

We found that although there are no significant changes in the fold change expression of the SASP receptors (Figure 5—figure supplement 1 in the revised SI), there is a very significant increase in the fraction of the macrophages and neutrophils that express SASP receptors. These cell types are among the main mediators of the removal of senescent cells. Again, we would like to stress that we did not cherry-pick the genes and took into account the global change. The false discovery rate of macrophages changing the fraction of their SASP milieu is slim.

Finally, following the reviewer’s comments, we also fine-tuned our statements regarding the SASP and revised Figure 6, accordingly.

15. Throughout the paper, it is important to show (by either experiments or using publicly available resources) that the changes observed are indeed specific to the ovaries.

In the original manuscript, we have shown that the abundance of the most frequent ovarian cell types (DNT in the old females and ILC1 in the young ones) is ovary specific. We have shown that the abundance of the double negative T cell population of old females is ovary-specific (Figure 1—figure supplement 7) compared with the spleen and the peritoneum. We have also shown that the abundance of the ILC1 population in young females is ovary specific compared with the spleen (Figure 1—figure supplement 5).

Following the reviewer’s comment, we performed additional experiments that also show that the *change* in CD3^+^ lymphocytes (and its subpopulations, DNT+NKT, CD4, and CD8) composition is ovary specific (Figure 2—figure supplement 4).

We also revisited the current literature (see our response to comment 21 below), there is no evidence for the presence of DNT cells in other tissues such as the bladder, lung, and kidney.

16. The authors start the paper by discussing the reduced fertility as a function of age, so any of these results suggest a mechanisms for that? Some discussion of this point will be useful.

Following the reviewers’ comment, we added the following addition to the discussion of the revised manuscript.

Age-related disruption of tightly controlled immune functions may lead to fertility decline. Turner et al., show that macrophages are pivotal in maintaining ovarian vascular integrity as their progressive ablation results in ovarian hemorrhage and tissue structural damage (Turner et al., Reproduction, 2011). Our results indicate a decline in the macrophage population with age which may result in fertility deterioration because of impaired ovarian vasculature. In addition, enrichment in CD3^+^ lymphocytes in the ovary was found to be associated with poor follicular reserve (Jasti et al., Biology of Reproduction, 2012) and thus may have an impact on ovarian autoimmune diseases. Moreover, one of the characteristics of polycystic ovary syndrome (PCOS) is an abnormal T lymphocyte milieu that is suspected to be involved in disease pathogenesis and ovarian dysfunction that leads to infertility (Li et al., Scientific Reports, 2019). Similarly, our results indicate an age-dependent change in the T lymphocyte population that may be involved in improper immune regulation in the ovary and hence lead to infertility. More specifically, our results show a decreased expression of several members of the TGF superfamily, such as *Inhba* and Acvr2a, by macrophages at old age. A previous study has shown that Activin A (which is composed of two Inhibin-ab, the translation product of *Inhba*), is related to LH secretion, plays a significant role in expanding the primordial follicle pool, and contributes to the early stages of follicular growth.

17. Because the scRNA-seq data presented by the authors show that the CD4^-^ CD8^-^ double-negative T cell subset co-express Trbc2 (TCRb) and Tcrgc2 (TCRg) genes, it would be important to test if these cells also co-express TCRb and TCRg/d at the protein levels. Pro-inflammatory CD4^-^ CD8^-^ double-negative T cells co-expressing TCRb and TCRg/d have been found in mice (Edwards et al., J Ex Med 2020), and it would be interesting to test whether the ovarian DNT cells show phenotypical or functional similarities with this cell type.

As the reviewer mentioned, Edwards et al., identified a unique population of hybrid αβ-γδ T cells in human PBMC and mouse spleen and LNs.

Following the reviewer’s comment, we performed additional experiments to test this point. First, we have verified our ability to observe γδ T-cells in our model. FACS measurements of mouse spleen samples show that the TCRγδ DNT cells (CD3^+^ CD4^-^ CD8^-^) constitute 48% and 23% out of the total DNT population for young and old mice, respectively. However, in the ovaries, the fraction of γδ DNT out of the overall DNT cells is low ~2.5% and age-independent (Figure 1—figure supplement 8). These results suggest that while a subset of the T-cells co-express Trbc2 and Tcrgc2, at the protein level the population that is TCRγδ DNT (which is the upper limit for the TCRγδ TCRαβ DNT) is only a small fraction (~2.5%, an order of magnitude smaller than the spleen) out of the total DNT.

18. To better understand the function of double-negative T cell subset in aging ovaries, one possible way would be to purify these cells and measure which cytokines they produce after TCR activation in vitro and/or co-culture these cells with activated CD4/CD8 T cells in vitro to test if they are capable of suppressing T cell proliferation.

We thank the reviewer for the suggestion. The goal of this manuscript is to reveal the immune milieu in the ovaries and their modulation due to maternal aging and analyze them. Performing such an experiment can be an excellent first step in exploring the possible direct mechanistic interaction between ovarian DNT and CD4/CD8 T cells. Yet, the number of immune cells that can be recovered from the ovaries is small (~20,000-50,000 CD45^+^ cells per mouse) and the number of DNT cells is even smaller (~1,000 cells per young mouse and ~15,000 cells per old mouse). It will require tens of mice to perform such a co-culture or to harness methods such as microfluidic chambers or wells (to perform co-culturing in smaller volumes). While deciphering the mechanics of the DNT population is an interesting follow-up, these types of in-vitro experiments are outside the scope of this paper.

19. For the cluster annotation of scRNA-seq data, it would be interesting to perform additional gene expression analyses to test whether the two clusters of dendritic cells correspond to cDC1 and cDC2 populations.

We thank the reviewer for this insightful comment – this is indeed the case. We tested the expression levels of several literature-based markers for cDC1 and cDC2 and show that cluster 3 corresponds to an expression pattern of cDC2, while cluster 4 corresponds to an expression pattern of cDC1. Figure 1—figure supplement 6 in the revised supplementary information illustrates this analysis.

20. Flow cytometry validation of scRNA-seq data in larger groups of mice presented in this study is chiefly limited by CD3^+^ T cells and CD11b+ cells. Additional flow cytometry experiments that validate alterations of central ovarian immune cell populations in old competed to adult mice would be helpful. Gating strategies for all flow cytometry experiments should be shown.

Following the reviewer’s comments, we have performed additional flow cytometry experiments. We were able to break down the CD3^+^ population into its subpopulations and measured the change in NKT cells, CD4, CD8, and double negative T-cells (Figure 2D). In addition, we also validated the changes in the fractions of several innate immune populations such as dendritic cells, neutrophils, and group-1 ILC (ILC1 and NK cells). These results are shown in Figure 2—figure supplement 2 in the revised supplementary information. Our gating strategies are shown in the SI (Figure 1—figure supplement 4).

21. It would be interesting to compare the scRNA-seq data generated by the authors with published datasets on the immune aging in various mouse tissue (e.g., Almanzar et al., Nature 2019; Kimmel et al., Genome Res 2019; Mogilenko et al., Immunity 2021) to identify common and tissue-specific immune changes in aging ovaries.

We thank the author for this important comment. These recent studies reveal age-specific changes in cell populations in different mouse tissues, independently of female fertility-related aging. Our data compare between young and aged female mice with an emphasis on their fertility window. Aged mice at that age do not necessarily exhibit an aging phenotype in other tissues, which are fertility non-related. As our study’s most aged group is 40-47 weeks old (~9-11 months) we can deduce that female mice of much older age such as 18m or 21m old (Almanzar et al.), 22-23m old (Kimmel at al. – males only) and 17-24m old (Mogilenko et al.) are post-estropausal.

Interestingly, we reveal that the ovarian immune milieu is characterized by age-related decrease in inflammatory hallmarks confined to the female fertile window period, while Almanzar et al., Kimmel at al. and Mogilenko et al., report increased expression of inflammatory markers at older ages. For example, bladder cells from both male and virgin female mice exhibit a shift towards a pro-inflammatory microenvironment when comparing mice at 1, 3, 18 and 24 months. This shift is reflected by an increase in tissue leukocytes and pro-inflammatory markers expression, together with a decrease in anti-inflammatory markers expression (Almanzar et al.). Age-related upregulation of pro-inflammatory environment was observed in kidney, lung, and spleen tissues of male mice in both immune and non-immune cell types (Kimmel et al.). In addition, pro-inflammatory cytokines and chemokines were elevated in the serum of old male mice, along with an increase in the fraction of inflammatory *GzmK^+^* CD8^+^ T cells in all tested tissues (Mogilenko et al.).

While Almanzar et al., observed an age-related decrease in T cell population together with a B cell increase in both spleen and mammary tissues, our study of younger aged mice shows an opposite trend in the ovary (i.e., an increase in T-cell population in parallel to a decrease in B cells). Moreover, T cells in our study (most notably DNT cells) are the most prominent cell type in the aged ovaries.

Kimmel et al. compared cells harvested from old mice (22-23 months) with aged adult mice (7-8 months), the latter are age-equivalent to our old mice. The authors describe an increase in the prevalence of immune cells in old tissues, a phenomenon we also witnessed in the old ovaries. In addition, lymphocytes were found to be more abundant in lungs and kidneys at old age – consistent with our results in the ovaries. On the other hand, Kimmel et al., also report an increase in inflammatory properties, as reflected in GO enrichment analysis, which reveals an up-regulation of inflammatory processes in more than five cell types. This result, as already mentioned, is opposed to our findings, in which the inflammatory properties of the ovaries decrease, as the female mice approach the end of the fertile period.

In the study of Mogilenko et al., the most prominent effect found is the elevation in *GzmK^+^* CD8^+^ T cells in the spleen, lungs, liver, and peritoneum. In our data, the fraction of these cells is insignificant (less than 1%) and decreases 1.6-fold in older mice.

The findings observed by these papers are consistent with the previously defined inflammaging, in which aging is accompanied by the development of a pro-inflammatory environment. It is important to note that our study compares female mice during their fertile period, which could explain the absence of inflammaging in the ovaries at older age. Furthermore, our study suggests that ovarian immune aging is not linear, but a complicated process that exhibits alternations between anti- and pro-inflammatory environment.

Following the reviewer’s comment, we added this discussion to the ‘Introduction’ and ‘Discussion’ sections of the revised manuscript.

22. Predicted changes in cytokine and chemokine expression levels and the crosstalk between immune and senescent cells presented in this study are based on scRNA-seq data but are lacking additional validation. For example, protein-level confirmation for some of these pathways would add important information about the mechanism of immune aging in the ovaries.

We thank the reviewer for this important comment. To validate our connectome results even further, we first performed additional computational analysis using a tool (Cell2Cell) developed by another group (Figure 4—figure supplement 1A). The results of this analysis were consistent with our own analysis, pointing CCR5 as the central hub of interaction that is decreased due to maternal age.

We used flow cytometry to test this result directly and show that, in agreement with our analysis, the fraction of cells that express CCR5 indeed decreases significantly (Figure 4—figure supplement 1B).

23. In Methods: antibody clone 17A2 is used for CD3 and CD4 detection (possible mistake).

We thank the reviewer for this correction. We have corrected the typo for CD4 clone that should be GK1.5.

[Editors' note: further revisions were suggested prior to acceptance, as described below.]

The manuscript has been improved but there are some remaining issues that need to be addressed, as outlined below:Reviewer #1 (Recommendations for the authors):Overall, while the authors have extensively addressed most of my concerns regarding the compositional analysis, the claims around a decreased inflammatory state with aging (particularly Figure 4B and the response regarding GO terms including both positive and negative regulators), and cell-cell communication analysis. I find the distinction between Figure 5 and Figure 5—figure supplement 1 to be interesting, with SASP recognition seemingly affecting a larger fraction of macrophages but not the average expression between the conditions. I also find it interesting that this same cell type decreases in abundance with age, possibly indicating that a subpopulation of macrophages that are retained with age are those exhibiting SASP recognition as an alternative explanation to the more natural conclusion that macrophages overall increase SASP recognition over time. While I am excited about this work, there are still some outstanding concerns, that I present below.– Were the log-normalized data scaled prior to dimensionality reduction? PCA typically takes scaled data as input.

Yes, the log-normalized data was scaled prior to dimensionality reduction (using Seurat scale.data). We have clarified it in the Cell type annotation section of the Methods in the revised MS.

– Reading through the methods, it is unclear whether the p-values used in DE testing were multiple tests corrected. Line 364 of the Related Manuscript File "100652_1_related_ms_2716393_rmry4f.pdf" states "all other" statistical analyses applied an FDR correction, implying that this wasn't applied to the DE and other statistical tests discussed in the "Statistical analyses" subsection of the Methods. Furthermore, in the Supplementary Table reporting DE results, there is only the column "pVal" indicating that this is not a multiple test corrected significance value. If multiple test correction was not applied to differential expression output p-values, they must be. Similar concerns for the GO enrichment results and MILO results, which also include many tests. Furthermore, given the LFC threshold filter, I wouldn't expect results to change drastically. However, if there are similar concerns to my original comments regarding how batch correction will affect downstream effect sizes, a demonstration that applying the multiple test correction does not change the results is a necessary minimum. I would suggest demonstrating that the genes identified as significantly differentially expressed with multiple test correction (perhaps with FDR ≤ 0.1) are consistent with those in the current list. This could be done by showing that the gene list sizes are similar and have a high Jaccard index.

We thank the reviewer for this very important comment. To remove any concerns regarding the rigor of our results, we have redefined what significant DEGs are. Following the reviewer’s comments, we took into account multiple hypothesis testing by applying the Storey q-values approach. As a cutoff, we used FDR≤0.1 (together with previous fold-change conditions). We have revised all the downstream analyses and used the new definition for significant DEGs for the GO analysis (Figure 3) and the connectome analysis (Figure 4).

While, as expected, fewer genes are now considered significant, our conclusions still hold both in terms of biological processes and in terms of chemokine and cytokine interactions. Most importantly, our conclusion regarding the chemokine and cytokine changes due to age does not change. The chemokine network exhibits the same downregulation profile, with a downregulation of CCR2 and CCR5 (which are consistent with our experimental validation of the decreased expression of CCR5 in dendritic cells). In the cytokines network, only three receptors that were marginally significant, do not pass the FDR. Yet there is a significant downregulation in the IL-1 superfamily and in *Tnfsfr1b* and its ligand *Tnf* (TNFα).

Overall, the reviewer comments have strengthened the manuscript demonstrating the significance of our results.